# Improved Skin Barrier Function Along with Hydration Benefits of *Viola yedoensis* Extract, Aesculin, and Schaftoside and LC-HRMS/MS Dereplication of Its Bio-Active Components

**DOI:** 10.3390/ijms252312770

**Published:** 2024-11-27

**Authors:** Sreelatha Thonthula, Sandra De Sousa, Alexis Dubuis, Samia Boudah, Richa Mehta, Akanksha Singh, Joan Eilstein, Jean-Claude Tabet, Sherluck John, Dhimoy Roy, Steve Thomas Pannakal

**Affiliations:** 1L’Oréal Research and Innovation, Bangalore 560067, India; sreelatha.thonthula@loreal.com (S.T.); richa.mehta@loreal.com (R.M.); akanksha.singh@loreal.com (A.S.); sherluck.john@loreal.com (S.J.); 2L’Oréal Research and Innovation, 93600 Aulnay-Sous-Bois, France; sandra.desousa@loreal.com (S.D.S.); samia.boudah@loreal.com (S.B.); joan.eilstein@loreal.com (J.E.); 3Faculty of Sciences and Engineering, Institut Parisien de Chimie Moléculaire, Sorbonne University, 75005 Paris, France; jean-claude.tabet@sorbonne-universite.fr; 4Medicines and Health Technologies Department (DMTS), CEA, INRAE, MetaboHUB, Paris-Saclay University, 91190 Gif sur Yvette, France; 5L’Oréal Research and Innovation, Mumbai Maharashtra 410210, India; dhimoy.roy@loreal.com

**Keywords:** *Viola yedoensis*, roseoside, platanionoside B, eriojaposide B isomer, calenduloside F, skin barrier, hydration, structural elucidation, LC-HRMS

## Abstract

The skin hydration level is a key factor that influences the physical and mechanical properties of the skin. The stratum corneum (SC), the outermost layer of the epidermis, is responsible for the skin’s barrier function. In this study, we investigated the role of a unique composition of *Viola yedoensis* extract for its ability to activate CD44, a cell-surface receptor of hyaluronic acid, and aquaporin-3, a water-transporting protein, in human keratinocytes (HaCaT). An ELISA assay evaluating the protein expression levels of CD44, aquaporin-3 (AQP3), filaggrin, and keratin-10 revealed that *V. yedoensis* extract upregulated the levels of CD44 and AQP3 by 15% and 78%, respectively. Additionally, *V. yedoensis* extract demonstrated a comparative effect on water vapor flux in TEWL and lipid perturbation in DSC versus the reference, glycerin. In light of this new biological efficacy, a detailed phytochemical characterization was undertaken using an integrated LC-HRMS/MS-based metabolomics approach, which provided further insights on the chemistry of *V. yedoensis*. This led to the identification of 29 secondary metabolites, 14 of which are reported here for the first time, including esculetin, aesculin, apigenin and kaempferol C-glycosides, megastigmane glycosides, roseoside, platanionoside B, and an eriojaposide B isomer, along with the rare, calenduloside F and esculetin diglucoside, which are reported for the first time from the genus, *Viola*. Notably, two active components identified in the *V. yedoensis* extract, namely, aesculin and schaftoside, showed an upregulation of the protein expression of CD44 in HaCaT cells by 123% and 193% within 24 h of treatment, respectively, while aesculin increased AQP3 levels by 46%. Aesculin and schaftoside also significantly upregulated the expression of K-10 levels by 299% and 116%, which was considerably higher than sodium hyaluronate, the positive control. The rationale used to characterize the new structures is outlined along with the related biosynthetic pathways envisioned to generate roseoside and Eriojaposide B. These findings provide new molecular insights to deepen the understanding of how *V. yedoensis* extract, along with the biomarkers aesculin and schaftoside, restores the skin barrier and skin hydration benefits.

## 1. Introduction

The skin, our body’s largest interfacial film, acts as a crucial barrier separating the human body from the external environment [1,2]. Primarily composed of stratum corneum, it consists predominantly of keratin and lipids, forming a physical layer that serves as a barrier against trans-epidermal water loss (TEWL) and pathogens. Meanwhile, the lipids and natural moisturizing factors within this layer contribute to its chemical barrier properties [3,4]. Furthermore, the stratum corneum is structured with multiple levels of corneocytes embedded in a lipid matrix comprising ceramides, cholesterol, and fatty acids. These lipids are distributed across two layers, known as the stratum disjunctum and stratum compactum. The cornified cell envelope (CE) present on the outer surface of the epidermis is a part of the skin barrier. The primary constituents of the CE are CE keratins, making up 85% of differentiated keratinocytes. The cornified cell envelope is made of proteins such as involucrin, loricrin, and proline-rich proteins such as SPRR [5]. The upregulation of these proteins help in providing an adaptive response of the epidermis to UV radiation by maintaining the CE barrier function. Loricrin and involucrin are structural proteins crucial for skin barrier function and integrity [6]. Filaggrin, a filament aggregating protein is an important component of the cornified cell envelope. Produced as the precursor proprotein, profilaggrin, it is found in the keratohyalin granules of the granular layer. As the granular cells differentiate into cornified cells, profilaggrin is dephosphorylated and cleaved by several endoproteases to generate filaggrin monomers–amino acids and their derivatives. The filaggrin breakdown products form the natural moisturizing factors (NMFs), which contribute to epidermal hydration and barrier function [7]. This intricate composition reinforces the skin’s protective function [8]. The retention of water in the SC is dependent on the presence of natural hygroscopic agents within the corneocytes, collectively referred to as natural moisturizing factors or NMFs [9,10]. Glycerol, an endogenous humectant derived from sebaceous glands, along with humectants like glycerol, draws water from the dermis to the SC and upregulates the expression of aquaporin channels, claudins, and filaggrin, thereby increasing hydration and preventing water loss [11]. Aquaporins constitute a family of transmembrane channels responsible for transporting water and, in certain instances, small solutes across the plasma membrane through osmotic gradients. Among these, aquaporin-3 (AQP3) stands out as the predominant and extensively studied aquaporin functioning as an aquaglyceroporin, demonstrating its capability to transport water, glycerol, urea, and hydrogen peroxide [12].

Another important component of our skin is hyaluronic acid (HA), a natural, non-sulfated, non-branched glycosaminoglycan that is a major component of the dermis. HA is also reported to be present in the epidermis and interacts with CD44, which functions as a coupling site on keratinocyte cell surface, thereby regulating keratinocyte differentiation and SC-extracellular lipid formation [13]. CD44 is associated with the regulation of keratinocyte proliferation in response to extracellular stimuli and the maintenance of local HA homeostasis [14]. Concurrently, aquaporins, filaggrin, and CD44 contribute to the maintenance of moisture and lipid content in the skin. The differentiation of keratinocytes from the basal to the spinous layer is characterized by a shift in keratin production from keratins 5 and 14 to keratins 1 and 10. In addition to keratins 1 and 10, the cells in the spinous layer also synthesize involucrin. As cells mature, they transition to the granular layer, expressing profilaggrin and loricrin [15]. These proteins, along with the lipids synthesized by these cells, form the cornified layer, which is an essential part of the epidermal barrier, preventing fluid loss and protecting the organism against invasion by pathogens.

Over the past decade, there has been a notable surge in interest among researchers, the cosmetic industry, and consumers worldwide for botanical ingredients and their secondary metabolites sourced from natural origins. This increased attention is driven by the perception that these natural compounds are not only effective but also safer, non-toxic, and free from adverse side effects. The market for natural cosmetic products is experiencing rapid growth and is considered one of the fastest-growing sectors globally. Projections indicate that it is poised to achieve a substantial value, reaching approximately USD 30.1 billion by the year 2026 [16]. This reflects the increasing consumer preference for natural and sustainable beauty solutions, driving the expansion of the natural cosmetics market. Cosmetics enriched with bioactive compounds are considered well suited to meet the specific needs of the skin, and they are often viewed as more environmentally friendly compared to conventional cosmetics. In recent years, there has been a growing interest in dermocosmetics and cosmeceuticals derived from plant materials. This heightened attention reflects an increased focus on plant-based products known for their beneficial properties in skincare [17]. The trend reflects a broader consumer interest in natural and botanical solutions within the realm of skincare and cosmetic formulations.

In recent years, there has been renewed focus on investigating plants for their potential benefits in skin hydration. Literature insights reveal *Aloe barbadensis*, commonly known as *Aloe vera*, yields a gel extracted from its succulent leaves. The gel acts as a humectant and attracts water from the dermis below, aiding in maintaining hydration by binding this water within the stratum corneum [18]. *Saccharum officinarum* (sugarcane) contains starches and sugars rich in potassium and dimethyl sulfoniopropionate (DMSP) that have skin hydrating properties [19]. However, despite this potential, a limited number of natural products have been thoroughly investigated for their benefits regarding skin hydration and epidermal barrier restoration. This gap in research prompted our exploration of plants within the genus *Viola* for their potential hydration and skin restoration benefits. To date, there have been several prior investigations on two related *Viola* species, *Viola tricolor* and *Viola*, which have been an occasional source of flavone glycosides and coumarins [20,21]. Our attention was drawn to *Viola yedoensis* Makino or Chinese violet, a well-known traditional Chinese medicinal plant because its extract was potently active in a primary screen for hydration benefits. *V. yedoensis* is reported to be used to treat various skin disorders such as allergic dermatitis [22], hyperplasia, and edema. The plant is a rich source of flavone glycosides [23], coumarins [24], and phenolic acids, which are known for their diverse bioactivities. Building on this foundation, a flavonoid-enriched extract of *V. yedoensis* was developed and evaluated for its effects on skin hydration and barrier function. Thus, the whole plant of *V. yedoensis* was subjected to a one step hydroalcoholic extraction, and the extract enriched with flavonoid glycosides was evaluated in the enzyme-linked immunosorbent assay (ELISA) against the keratinocyte differentiation markers CD44, involucrin, filaggrin, keratin-10, and aquaporin-3 with sodium hyaluronate as the positive reference in HaCaT cells. The extract enriched with the flavonoid glycosides was fully characterized using liquid chromatography coupled to high-resolution tandem mass spectrometry (LC-HRMS/MS). Dereplication with product ion spectrum databases, divulging the fragmentation pathways of deprotonated glycoconjugates and the interpretation of dissociation mechanisms, was proposed to annotate 29 secondary metabolites, 14 of them being identified for the first time for the genus *Viola.* Additionally, an evaluation of the trans-epidermal water loss (TEWL, directly proportional to skin hydration in healthy skin) and differential scanning calorimetry (DSC, reference method for characterizing free water molecule inside SC) analysis of the extract was performed and compared with the reference glycerol and water. Finally, the major metabolites, aesculin and schaftoside, were evaluated for their in vitro properties on epidermal markers and on improving the skin hydration properties as well.

## 2. Results

### 2.1. Epidermal Barrier Function and Skin Hydration Evaluation of V. yedoensis Extract

The *V. yedoensis* extract was evaluated for the protein expression levels of CD44, aquaporin-3 (AQP3), keratin-10 (K-10), filaggrin (FLG), and involucrin (INVL) in HaCaT cells by sandwich-ELISA assay. A cytotoxicity assessment of *V. yedoensis* extract was performed, and the non-cytotoxic dose was evaluated for the modulation of AQP3 and CD44 markers.

### 2.2. Cell Viability Assay

To investigate the cytotoxicity of *V. yedoensis* extract in HaCaT cells, we conducted viability assays using a cell-counting kit. The immortalized human keratinocyte cell line HaCaT was used as a model to study of keratinocyte cytotoxicity. HaCaT cells were seeded in 24-well plates (Thermo Fisher Scientific, Carlsbad, CA, USA) at a density of 10,000 cells/cm^2^. Cell culture monolayers in 24-well plates were treated with *V. yedoensis* extract, ranging in concentration between 0.02 and 2.0%, which were applied after 14 days. All solutions were diluted in DPBS. According to our results, the IC_50_ (cytotoxic concentration) of *V. yedoensis* extract was found to be 0.07%. In contrast, treatments with 0.03% did not significantly affect cell viability, which remained close to the untreated control in the HaCaT cell type; therefore, we chose a 0.03% concentration as our test concentration for the study on the epidermal differentiation of markers. Overall, the cells looked visually healthy, and no cell peeling was observed. Sodium hyaluronate (5 kDa, 0.1%) was used as the positive control (Figure 1).

#### In-Vitro Study of Epidermal Differentiation Markers

The results of the treatment of HaCaT cells with *V. yedoensis* extract and the positive control, sodium hyaluronate, at 0.1% displayed an expression of epidermal differentiation markers, CD44, AQP3, and K-10, and are shown in Figure 2. The results of the ELISA assay indicate that our *V. yedoensis* extract and the low molecular weight hyaluronic acid sodium salt (5 kDa) were able to stimulate the protein expression levels of CD44 expression in HaCaT cell line by 15% and 86%, respectively. The CD44 measurement was normalized to total protein, so it shows that the increase in CD44 was due to an increase in expression (more CD44 per μg protein) and not due to an increase in the number of cells (no change in CD44 per μg protein). This information is important as it suggests that the increase in CD44 protein is not related to keratinocyte cellular proliferation but to actual protein expression increase. Once again, human keratinocytes (HaCaTs) were used in this study to examine the effect of *V. yedoensis* extract and the positive control on AQP3 expression because the expressions of cytokine/chemokine receptors are more stable in HaCaT than in NHKs. Our *V. yedoensis* extract upregulated the protein expression of AQP3 in HaCaT keratinocytes cells by 78%, which was significantly higher than the positive reference at 54% within 24 h treatment, respectively (Figure 2), suggesting that *V. yedoensis* extract may facilitate the transport and metabolism of water and glycerol in the skin epidermis and the function of the epidermal water permeability barrier. Previous reports suggested that the overexpression of Akt could at least partially contribute to the increase in AQP3 and consequently lead to autophagic responses, which means P13K/AKT could mainly control the upregulation of AQP3. However, we did not examine the P13/AKT pathway and confirm the same in our study. There are possibilities that Ras, ERK, and AMPK could be mediated to induce AQP-3, which we plan to examine in the future. To conclude, *V. yedoensis* extract did not upregulate the expression of FLG, INVL, and K-10 in HaCaT cells. All values were normalized with total protein yield.

### 2.3. Skin Hydration Evaluation

TEWL analysis: In normal conditions, with an intact skin barrier, TEWL on human skin is on the order of 5–10 g/m^2^/h. The TEWL measurements with isolated stratum corneum treated by *V. yedoensis* extract were comparable to water and 5% glycerin in water. It has a similar hydration effect as 5% glycerin (Table 1).

DSC analysis: The thermogram of the untreated stratum corneum is characterized by four phase transitions—T1 (40 °C), T2 (72 °C), T3 (85 °C), T4 (>110 °C). The endothermic peaks corresponding to transitions T1, T2, and T3 for a typical stratum corneum sample are shown in Figure 3. *V. yedoensis* extract shows a lipid perturbation of ±0.30 °C in DSC analysis (Table 1).

### 2.4. Chemical Characterization of the Viola yedoensis Extract

#### 2.4.1. Methodology

The *V. yedoensis* extract was fractionated and chemically characterized by LC-V/HRMS. This fractionation aimed to enrich minor secondary metabolites within the extract. Spectral comparison with databases and the interpretation of dissociation mechanisms led to the identification of 29 secondary metabolites, 14 of which are reported here for the first time in *V. yedoensis*. Figure 4 presents the methodology and data used to annotate both known and novel compounds from *V. yedoensis* extract.

Of the 29 compounds identified, 12 were annotated with the highest level of confidence by comparing raw data, collisional spectra (conditions in experimental part), and retention time with authentic compounds (10 were already reported in *V. yedoensis*). Putative identifications were proposed for 6 compounds based on MS/MS spectral matching with databases (without RT confirmation), and 11 compounds were tentatively identified through MS/MS spectra interpretation (of which, 5 were already reported in *V. yedoensis*).

To do so, stable ionized molecular species are essential to provide specific structural information that can be used for the elucidation of unknown metabolites or to propose putative structures. As polar molecules, glycoconjugates require electrospray ionization (ESI) conditions to be desorbed as stable even-electron molecular species [26]. Due to the presence of OH groups, numerous acidic sites are located on both aglycones and carbohydrates. Consequently, the negative ionization mode was chosen to generate deprotonated molecules. Under these conditions, a mixture of competitive deprotomers arises from desolvation of charged aggregates during the ESI process. The negative charge of these activated deprotomers probably favors different fragmentation orientations, providing complementary structural information using high-resolution tandem mass spectrometry. Notably, unlike protonated molecules where C-C and C-O aglycon linkages lead to cross-ring dissociations of hexose/pentose moiety, only deprotonated molecules with a C-C bond to the aglycone lead to such cleavages, whereas those with a C-O bond led to the ether cleavage yielding a dehydrated carbohydrate release. This distinction provides a reliable way to differentiate between O-aglycon and C-aglycon linkage with sugars. Furthermore, to obtain diverse structural information such as carbohydrate location on the aglycon and sugar linkage patterns, non-resonant ion excitation was employed to enable the construction of informative MS/MS spectra databases, consisting of known compounds. These databases facilitated the interpretation of deprotonated molecule fragmentations, leading to the establishment of specific decomposition rules. These observations were then applied to propose putative structures for unknown compounds. To distinguish the location of different hexose and pentose moieties on the aglycone skeleton, product ion spectra obtained in resonant excitation mode were used. This mode mainly yields competitive fragmentations directly from the selected precursor ion, which results in the first generation of product ions [27]. To illustrate this approach, flavonoid and terpenoid glycoconjugates within *V. yedoensis* extract were selected to propose putative structures and highlight fragmentation rules applicable to the structural elucidation of other natural products.

#### 2.4.2. Identification of Known Compounds from *Viola yedoensis*

Figure 5 shows LC-UV/HRMS chromatograms of *V. yedoensis* extract, including (a) the UV maxplot and (b, c) total ion current (TIC) acquired in positive and negative ion modes, respectively.

UV maxplot and TIC profiles exhibit a high degree of superposition, indicating that major UV-absorbing components were also detectable by mass spectrometry. This observation confirms the suitability of HRMS/MS for their structural characterization. Notably, positive and negative ionization modes demonstrated comparable efficiency for these compounds. Many of the observed signals were compounds previously reported from *V. yedoensis*. Their spectral characteristics are summarized in Table 2.

In Table 2, identified compounds from *V. yedoensis* extract already known from the plant belongs to coumarins (e.g., **3** and **10**), coumarin glycosides (e.g., **1**, **2**, and **7**), flavonoids (e.g., **15**), and flavonoid mono- and di-glycosides (e.g., **4**, **5**, **6**, **8**, **9**, **11**, **12**, **13**, and **14**). Most of the flavonoids are O-glycoside and C-glycoside forms of apigenin (flavone), luteolin (flavone), and kaempferol (flavonol). Among the coumarins, aesculin (**2**) was identified as the main compound, while schaftoside (**6**) was found to be the major C-glycoside flavone in *V. yedoensis* extract, considering all the modes of detection (see Figure 4). Because of their abundance in *V. yedoensis* extract, aesculin (**2**) and schaftoside (**6**) were later studied in vitro on epidermal differentiation markers and for their skin hydration benefits.

#### 2.4.3. Confirming Known and Elucidating New Bioactive Structures in *Viola yedoensis*

##### Flavonoid Glycoconjugates

Fragmentation pathways of deprotonated glycoconjugates are elucidated by considering the mobile proton theory or charge delocalization through overlapping orbitals. When these processes are hindered by steric constraints or significant spatial separation between the charge site and the bond involved for cleavage, deprotonated molecule isomerization into an ion/dipole complex must be considered [35]. This facilitates negative charge transfer between distant sites, driven by the relative gas-phase acidities of the participating functional groups (hydroxyl groups).

For instance, the estimated difference of gas-phase acidity between an OH group of aglycone and a dehydrated hexose is larger than 70 kJ/mole (details in experimental part). Consequently, proton transfer from the phenolic group can neutralize the deprotonated hexose (e.g., alkoxide site) that occurs since it is an exothermic pathway. Conversely, without a mobile proton on the aglycon, proton transfer is precluded. Consequently, negative charge transfer to the aglycon requires hydride transfer from the deprotonated carbohydrate within the ion/dipole complex, effectively reducing the quinone form of oxidized aglycon. This proposed mechanism is illustrated in Figure 6 with quercetin-3-sophoroside-7-O-rhamnoside (**21**). Under these conditions, fragmentations of deprotonated luteolin (15) are essentially limited to two competitive processes initiated by the heterocycle opening at the C3-C4 bond induced by a phenoxy site, followed by the cleavage of either the C2-C1’ linkage (the formation of *m/z* 175) or the C2-O bond (the formation of *m/z* 151) (Appendix A). Conversely, deprotonated glycoconjugates listed in Table 1 decompose primarily through the loss of a carbohydrate moiety (either a complete carbohydrate unit or a fragment thereof) or a glucuronic acid unit. This fragmentation pathway consistently yields the deprotonated aglycone. The specific fragmentation patterns observed are contingent upon the nature of the glycosidic linkage connecting the aglycone and the carbohydrate, either O-glycoside or C-glycoside.

In all the studied cases, the release of saccharides linked to aglycone involves benzylic C-O bond cleavage to regenerate intact deprotonated aglycone, as opposed to the significantly less favorable cleavage of phenylic bond. Furthermore, the benzylic cleavage must proceed through a stepwise mechanism involving the formation of an ion/dipole complex intermediate [36,37]. Such a mechanism would enable the migration of the negative charge if required for consecutive/competitive possible pathway(s).

This fragmentation behavior is exemplified by the deprotonated glycosidic linkage positional isomers as cichoriin [**1** − H]^−^ and aesculin [**2** − H]^−^, both containing a single hexose moiety. Under resonant excitation conditions (i.e., collision-induced dissociation, CID), the product ion spectra of these anions, acquired at a normalized collision energy (NCE) of 25% (E_lab_), consistently show a product ion at *m/z* 177, attributed to the dehydrated glucose (C_6_H_10_O_5_, 162 Da) loss (Appendix A). Notably, this fragmentation pathway is more prominent for deprotonated cichoriin compared to aesculin. This difference can be rationalized by considering the lower gas-phase acidity of the phenolic group in (**1**) relative to (**2**). As a result, the phenoxide anion in deprotonated (**2**) is stabilized by greater charge delocalization compared to its analog in deprotonated (**1**) illustrated in Appendix A. A plausible stepwise dissociation mechanism, illustrated in Appendix A, involves a common ion/dipole intermediate, subsequently losing C_6_H_10_O_5_ to form the common product ion at *m/z* 177. This rationalizes why an abundance of the survivor [**1** − H]^−^ precursor ion is lower than that of deprotonated aesculin, [**2** − H]^−^. A possible reactional pathway rationalizes the favorable dissociation of [**1** − H]^−^ compared to that of [**2** − H]^−^ where the charge is dispersed on two oxygen atoms (Appendix A).

Glycoconjugates **12** (Table 2), **18**, **27**, and **28** (Table 3) appeared similar to compounds **1** and **2** as they all contain a monosaccharide linked via a benzylic C–O bond and exhibit a loss of dehydrated glucose (162 Da) upon CID. This fragmentation follows a pathway analogous to that depicted in Appendix A. Similarly, compound **7**, containing an acetylated glucose moiety, undergoes analogous fragmentation with a neutral loss of 204 Da. The deprotonated [**23** − H]^−^ ion (*m/z* 433), featuring a rhamnose-O-aryl linkage, loses dehydrated rhamnose (146 Da). The loss of 162 Da is also observed in disaccharide conjugates, such as the [**20** − H]^−^ precursor ion (*m/z* 597), which contains one O-phenyl linkage. However, the fragmentation pattern of disaccharide conjugates is influenced by the aglycone structure. For instance, the [**17** − H]^−^ ion (*m/z* 471) can directly lose both linked carbohydrates (294 Da), yielding the deprotonated aglycone at *m/z* 177. This process likely proceeds through a stepwise mechanism involving isomerization to an ion/dipole complex prior to dissociation, similar to the mechanism proposed for the [**1** − H]^−^ and [**2** − H]^−^ isomer precursor ions in Appendix A. The presence of a phenol group ortho to the O-glycoside linkage appears to promote the direct loss of the disaccharide unit, whereas its absence allows for the consecutive losses of individual sugar.

Interestingly, a competing fragmentation pathway involving the loss of a radical species, formally C_6_H_11_O_5_^•^ (163 Da), is observed for monosaccharide compounds **13** and **14**. This radical loss, potentially corresponding to a 1-deoxyglucose or 1,5-anhydroglucitol radical, occurs in competition with the loss of 162 Da that corresponds to dehydrated glucose. The observation of this radical loss is noteworthy, as it seemingly breaks the even-electron rule [38]. While the exact mechanism driving this radical loss remains to be fully elucidated, it may involve either a direct loss of C_6_H_11_O_5_^•^ or a consecutive loss of C_6_H_10_O_5_ and H^•^. Notably, the position of the saccharide moiety on the aglycone skeleton and the location of phenolic groups appear to influence the prevalence of this radical anion formation. For instance, deprotonated quercetin-3-glucoside (**13**, 5,7,3′,4′-tetrahydroxyflavonol backbone) and astragalin (**14**, 5,7,4′-trihydroxyflavonol) preferentially undergo the 163 Da loss. In contrast, deprotonated isorhamnetin 3-O-glucoside (**27**, 5,7,4′-trihydroxy, 3′-O-methoxyflavonol) and cynaroside (**12**, 5,7,3′,4′-tetrahydroxluteolin), with the glycosidic linkage at C(7), favor the neutral loss of dehydrated glucose (162 Da). Further investigations are currently underway to elucidate the factors governing the competition between these two fragmentation pathways and will be reported elsewhere.

**Table 3 ijms-25-12770-t003:** Unreported compounds in *V. yedoensis* identified in extract with *m/z* of precursor ions [M − H]^─^ and [M + H]^+^ and their product ions using LC/ESI-HRMS/MS under resonant (CID 25 %) and non-resonant collisional (HCD 20%) excitation conditions.

Index	Rt (min)	Name/CAS Number (Identified with: ^a^ MS² spectra, ^b^ RT, ^c^ De Novo)	Glycosyl Linkage to Aglycon Backbone	Precursor Ion	*m/z* of Precursor Ion	δ ppm	Molecular Formula	Excitation Conditions	Product Ion *m/z* (^‡^ Oxidation)	References
16	5.39	Esculetin diglucoside ^c^	O-coumarin	[M − H]^−^	501.1239	1.1	C_21_H_26_O_14_	CID 25%	177.0192, 133.0289, 105.0339, 89.0390, 81.0339	[39]
17	5.58	5-hydroxy apiosylskimmin ^c^/1149372-95-1	O-coumarin	[M − H]^−^	471.1131	1.6	C_20_H_24_O_13_	HCD 20%	177.0188	[40]
18	6.59	Roseoside ^a,b^/54835-70-0	O-megastigmane	[M − H]^−^	385.1856	1.7	C_19_H_30_O_8_	CID 25%	223.1344, 205.1231, 161.0448, 153.0920	[41]
19	7.00	Meloside L ^a^/55196-48-0	C-flavone	[M − H]^−^	609.1485	4.8	C_27_H_30_O_16_	HCD 20%	489.1040, 447.0924, 429.0815, 399.0722, 369.0635, 357.0615	[42]
20	7.24	Quercetin 3-arabinofuranoside 7-glucoside ^c^/23394-51-6	Di-O-flavonol	[M − H]^−^ [M + H]^+^	595.1311 597.1452	0.6	C_26_H_28_O_16_	HCD 20%	463.0888 ^‡^, 433.0787, 301.0355 ^‡^, 465.1029, 435.0934, 303.0504 ^‡^	[43]
21	7.25	Quercetin-3-sophoroside-7-O-rhamnoside ^a,b^/64828-40-6	Di-O-flavonol	[M − H]^−^	771.1978	0.8	C_33_H_40_O_21_	CID 25%	625.1396, 446.0841 ^‡^, 301.0356 ^‡^	[44]
22	7.44	Kaempferol-3-sophoroside-7-rhamnoside ^a^/93098-79-4	Di-O-flavonol	[M − H]^−^ [M + H]^+^	755.1470 757.2203	1.6	C_33_H_40_O_20_	HCD 20%	609.1470, 431.0970 ^‡^, 285.0408 ^‡^, 433.1139, 287.0555	[45]
23	7.46	Kaempferol-7-O-rhamnoside ^a^/20196-89-8	O-flavonol	[M + H]^+^	433.1138	0.8	C_21_H_20_O_10_	HCD 20%	287.0554	[46]
24	8.33	Vicenin-1 6″-O-acetate ^c^/163345-99-1	Di-C-flavone	[M − H]^−^	605.1518	1.9	C_28_H_30_O_15_	HCD 20%	545.1312, 455.0990, 335.0570	[47]
25	8.64	Kaempferol 3-O-sophoroside ^c^/19895-95-5	O-flavonol	[M − H]^−^	609.1470	2.4	C_27_H_30_O_16_	HCD 20%	429.0834, 327.0512, 285.0407 ^‡^, 255.0300, 227.0352, 151.0036	[48]
26	8.66	Platanionoside B ^c^/23601-22-8	Di-O-megastigmane	[M − H]^−^	533.2608	1.9	C_25_H_42_O_12_	CID 25%	371.2072, 209.1534, 161.0448	[49]
27	9.54	Isorhamnetin 3-O-glucoside ^a^/5041-82-7	O-flavonol	[M − H]^−^	477.1045	2.5	C_22_H_22_O_12_	HCD 20%	357.0609, 315.0507 ^‡^, 285.0406, 271.0257, 243.0301	[30]
28	9.55	(4R)-4-[(3R)-3-[[3-O-(4-carboxy-3-hydroxy-3-methyl-1-oxobutyl)-β-d-glucopyranosyl]oxy]butyl]-3,5,5-trimethyl-2-cyclohexen-1-one ^a^/1039756-96-1	O-apocarotenoid	[M – H]^−^ [M + H]^+^	515.2484 517.2642	2.8 1.3	C_25_H_40_O_11_	CID 50% CID 25%	471.2578, 453.2473, 413.2162, 371.2059, 353.1954, 209.1539, 161.0438, 125.0235, 99.0443, 211.1691	[50]
29	12.04	Calenduloside F ^a^/51415-02-2	Di-O-triterpene	[M – H]^−^ [M + H]^+^	793.4389 795.4531	1.9 0.0	C_42_H_66_O_14_	CID 35% CID 20%	631.3845, 613.3723, 509.3618, 455.3509, 439.3578, 249.1848, 217.1950, 203.1792, 191.1794	[51]

Identified with authentic compound: ^a^ MS^2^ spectrum, ^b^ RT, and ^c^ de novo structural elucidation. ^‡^ Oxidation of aglycone to quinone-like structure characterized by high electron affinity yielding preferentially radical anion by electron transfer from deprotonated carbohydrate in ion/dipole complex (vide infra).

In the case of non-linked mono- and di-saccharides with a C-phenyl linkage, a strong hindrance of the dehydrated saccharide loss is expected due to charge-driven fragmentations. O-ring opening generates an alkoxide intermediate. This alkoxide facilitates 1,2 or 1,3 hydride transfers, leading to the release of C_2_H_4_O_2_ (60 Da loss), C_3_H_6_O_3_ (90 Da loss), or C_4_H_8_O_4_ (120 Da loss). Subsequent proton transfer from the neighboring phenolic site to the resulting allylic alkoxide can promote a second glycosidic ring opening, leading to further losses of 90 Da or 120 Da. However, in the case of linked disaccharides, such as iso-orientin 6’-O-glucopyranoside (**19**), the expected 162 Da loss is observed. This loss results from the cleavage of the glycosidic linkage between the hexose (or/and pentose) units without affecting the aglycone–hexose interaction. This cleavage pathway competes with the release of (CH_2_O)*_n_* units (*n* ≥ 2 from pentose linkages, as observed in compounds **9**, **24** and *n* ≥ 3 from hexose linkages, as in compounds **4**, **5**, **6**, **8**, **11**, and **19**) through consecutive cleavages within each pentose/hexose ring.

### 2.5. Identification and Characterization of Terpenoid Glycoconjugates

The fragmentation of both positively and negatively charged molecular species must be considered for the study of terpenoid glycoconjugates, as the dissociation provides complementary structural information. When a molecular species dissociates into two fragments, the charge distribution is determined by the gas-phase thermochemical properties of the corresponding neutrals of each fragment. Consequently, the charge-carrying fragment corresponds to the one who’s neutral is the strongest base in positive mode and the strongest acid in negative mode.

The protonated [**28** + H]^+^ molecule dissociates mainly to form *m/z* 211.1691 ion (C_13_H_23_O_2_, 2 ppm), which suggests that the protonated aglycon releases H_2_O (or 2 H_2_O) independently of the collision energy used for its activation. On the other hand, the collisional spectra of the deprotonated molecule [**28** − H]^−^ (*m/z* 515.2484, C_25_H_39_O_11_) recorded under several collision energy conditions display more product ions than observed from other deprotonated compounds, as recorded for deprotonated platanionoside B (**26**). The [**28** − H]^−^ and [**26** − H]^−^ precursor ions dissociate into a common fragment *m/z* 209.1539 ion (C_13_H_21_O_2_, 4 ppm), which is consistent with the *m/z* 211.1690 positive product ion (C_13_H_23_O_2_, 1 ppm). Notably, very weak abundant *m/z* 161.0457 corresponds to a dehydrated hexose C_6_H_9_O_5_ (4 ppm) formed with *m/z* 209.1539 from the competitive, stepwise dissociations of the *m/z* 371.2061 ion (a similar process is described in Appendix A). This further confirms that the aglycon of these compounds are similar, like β-ionone/β-ionol backbone. Two possible isomeric candidates can be considered initially, such as eriojaposide B and its apocarotenoid isomer. However, the former must be ruled out since the CO_2_ and (CO_2_ + C_3_H_6_O) losses cannot occur from hexoses or pentoses, unlike the apocarotenoid isomer with a 3-hydroxy-3-methylglutaryl end group and an ester linkage with a hexose. Indeed, from the latter, these losses can be explained by using processes promoted by the negative charge from the carboxylate end group. This CO_2_ release is assisted by proton migration from the neighboring hydroxy group in the 3-hydroxy-3-methylglutaryl end group, which allows consecutively the C_3_H_6_O release (1 ppm) as acetone neutral (*m/z* 413.2162, Appendix A). Alternatively, the complementary ion pair *m/z* 371.2061 (C_19_H_31_O_4_, 4 ppm) and *m/z* 143.0344 (C_6_H_7_O_4_, 3 ppm) can be rationalized by a stepwise dissociation of the deprotonated apocarotenoid isomer through isomerization into ion/dipole and internal proton transfer followed by the splitting of both partners, allowing the competitive formation of both fragment ions (Appendix A). This latter ion can (i) directly lose CO_2_ through the formation of a 5-membered lactone *m/z* 99 (Appendix A) or (ii) be isomerized into a hydroxylated cyclic anhydride with a negative charge carried by an enolate site of the intermediate (Appendix A). All these processes are consistent with a 3-hydroxy-3-methylglutaryl end group.

The positive mass spectrum of the compound **29** is characterized by a major ion at *m/z* 439.3578 (C_30_H_47_O_2_, 1 ppm), likely attributable to prompt dissociations occurring during charged aggregate desolvation. In contrast, the negative ion mode spectrum is dominated by a highly abundant signal at a higher *m/z* value of 793.4363, corresponding to an elemental composition of C_42_H_65_O_14_ (2 ppm error). Interestingly, a similar behavior is observed in tandem mass spectra of calenduloside E (retrieved from the mzCloud database) in positive mode at *m/z* 439.3567 (C_30_H_47_O_2_, 3 ppm) and in negative mode at *m/z* 631.3830 (C_36_H_55_O_9_, 2 ppm). These results suggest that both glycoconjugates share a common aglycone moiety, likely oleanolic acid, with calenduloside F (compound **29**) bearing an additional carbohydrate unit. Notably, product ions detected at *m/z* 249.1848 (C_16_H_25_O_2_, 0.5 ppm) and 191.1792 (C_14_H_23_, 2 ppm) generated from the selected ion *m/z* 439.3567 (C_30_H_47_O_2_, corresponding to dehydrated oleanolic acid) of compound **29** exhibit complementary elemental compositions relative to this precursor ion. These product ions are generated via molecular isomerization into ion/dipole intermediates that competitively undergo competitive fragmentations (Appendix A). Furthermore, a series of product ions in positive mode *m/z* 421, 393, 315, 217, and 203 are observed in collisional spectra of the prompt *m/z* 439.3567 fragment ion during MS/MS experiments of both protonated molecules. The formation of these product ions can be attributed to competitive, stepwise fragmentation processes.

### 2.6. Effect of Aesculin and Schaftoside on Epidermal Differentiation Markers In Vitro and Skin Hydration Benefits

Based on the ELISA assay results of *V. yedoensis* extract on epidermal differentiation markers, we were interested in decoding the active components responsible for the skin hydration benefits of this extract. Aesculin (coumarin glycoside) and schaftoside (apigenin-C-glycoside) were shortlisted to be evaluated for their effect on epidermal differentiation markers and hydration benefits based on their abundance in the extract, evident from the LC-HRMS/MS-based metabolomics approach under resonant (CID 25%) and non-resonant collisional (HCD 20%) excitation conditions as well as their wide range of biological activities previously reported, such as anti-inflammatory [52], wound healing [53], and protection against UVB-induced damage [3,54]. The two secondary metabolites were evaluated for the protein expression levels of CD44, AQP3, K-10, and FLG in HaCaT cells by sandwich-ELISA assay. A cytotoxicity assessment of aesculin and schaftoside was performed, and the non-cytotoxic dose was evaluated for the modulation of CD44, AQP-3, FLG, and K-10.

### 2.7. Cell Viability Assay

The half maximal inhibitory concentration, IC_50_ (cytotoxic concentration), of aesculin (0.5–2%) and schaftoside (0.05–0.2%) was performed to determine the non-cytotoxic concentrations to be used for the in vitro assay. The cell culture was maintained at 37 °C in a humidified incubator with 5% CO_2_ atmosphere and grown to 70% to 80% confluence. Then, 1% aesculin and 0.1% schaftoside were tested on HaCaT keratinocyte cells, and the percentage of HaCaT cell death observed was found to be <38% after 24 h treatment. Visually, the cells looked healthy, and no cell peeling was observed (Figure 7). Thus, 1% aesculin and 0.1% schaftoside were considered as non-cytotoxic to cells and tested in the ELISA assay at the same concentration. Overall, the cells looked visually healthy, and no cell peeling was observed. Sodium hyaluronate (5 kDa, 0.1%) was used as the positive control (Figure 7).

### 2.8. Epidermal Differentiation Markers In Vitro ELISA Assay

The two most abundant secondary metabolites of interest (aesculin and schaftoside) were added to HaCaT cells, and the AQP3 mRNA expression level was analyzed 24 h later. No morphological changes in cells were observed at a concentration of 1% and 0.1% for both the secondary metabolites. The coumarin glycoside, aesculin, significantly increased the mRNA expression level of AQP3 in the HaCaT cells compared to the control cells. Among them, the rate of increase in AQP3 levels induced by aesculin was significantly higher at 46%, closely similar to that of the positive control at a lower concentration (54% expression at 0.1%) and approximately 2-fold lower to that of the *V. yedoensis* extract (78% expression at 0.03%), suggesting that aesculin could be one of the main bioactive components responsible for the upregulation AQP3 levels in the *V. yedoensis* extract (Figure 1). These results suggest that aesculin contributes to an increase in AQP3 levels, which is important for the migration and proliferation of keratinocytes and the function of the epidermal water permeability barrier, by activating the P13K/Akt/mTOR signaling pathway, which we did not confirm in this study. Further, we observed that both aesculin and schaftoside upregulated the protein expression of CD44 in HaCaT cells by 123% and 193%, which was significantly superior to the positive control (86% at 0.1%) within 24 h treatment, respectively, suggesting that the topical application of the flavonoid, schaftoside, could provide both the biological benefits of CD44 stimulation and the physiological benefits of improved stratum corneum hydration that hyaluronic acid is popularly known to provide. In particular, the interaction between CD44 and HA is involved in keratinocyte differentiation and lipid synthesis. Additionally, the treatment of HaCaT keratinocytes with 1% aesculin, 0.1% schaftoside, and 0.1% sodium hyaluronate (positive control) upregulated the expression of K-10 levels by 299%, 116%, and 282%, respectively, suggesting that the two bioactive components could be responsible for maintaining epidermal integrity by increasing terminally differentiating epidermal keratinocyte differentiation. Interestingly, only schaftoside significantly upregulated the expression of filaggrin (FLG) by 93% (Figure 8), a structural protein that plays an important role in the skin’s barrier function and is implicated in some of the most common dermatological diseases, such as atopic dermatitis (AD) and ichthyosis vulgaris. Overall, we also observed that there was no upregulation in the involucrin levels on treatment with aesculin, schaftoside, or sodium hyaluronate.

## 3. Discussion

Skin hydration is a critical aspect of maintaining healthy skin, and water is generally considered essential for the normal functions of the skin [55]. Today, several cosmetic ingredients aim to restore physiological hydration while helping to decrease skin evaporation and improve the appearance and tactile properties of the skin [56]. Hyaluronic acid (HA), a glycosaminoglycan (GAG), provides hydration and structural integrity to the dermis. It has been reported that HA is also naturally present in the epidermis, binds to the extracellular space via CD44, and may play a role in epidermal barrier function and SC hydration [57]. In normal skin, HA is found in relatively high concentrations in the basal layer of the epidermis where proliferating keratinocytes are found [58]. Generally, CD44 is the main receptor for hyaluronic acid in the skin. Hyaluronic acid, the main ligand for CD44, binds to and activates CD44, resulting in the activation of cell signaling pathways that induce cell proliferation. A deficiency of CD44 leads to a reduction in HA, which in turn alters the proliferation and differentiation of keratinocytes, lipid synthesis, and ultimately the skin barrier function [13]. In our investigation, *V. yedoensis* extract stimulated the synthesis of hyaluronic acid in the skin endogenously by increasing the expression of CD44, the main receptor for hyaluronic acid in keratinocytes. Thus, it acts as a natural humectant, enabling skin hydration by binding and retaining water molecules in the epidermis. AQP3 is a member of the family of aquaglyceroporins, which are membrane proteins that form water channels across the cell membrane. They facilitate the transport of water and solutes like glycerol or urea [59]. AQP3 expression in SC is responsible for the function of water loss prevention. Both *V. yedoensis* extract and aesculin treatments resulted in the increased expression of AQP3 protein, suggesting enhanced AQP3 activity. Compounds that increase AQP3 in the skin are reported as skin moisturizers. Recent studies have demonstrated that ligands of either PPARγ or LXR increase the expression level of AQP3 in epidermal keratinocytes [60,61]. Furthermore, aesculin has been reported to suppress LPS-induced inflammation by activating PPAR-γ in RAW 264.7 cells (murine macrophage cell line) [62]. These results possibly suggest that aesculin may upregulate the expression of AQP3 via its PPARγ agonist action.

Filaggrin (FLG), a large protein localized in the corneocyte layer in the SC, is converted to NMF in the more superficial layers of the SC. Proteolysis of FLG is dependent on water activity within the corneocytes [63]. Interestingly, a previous study reported that apigenin significantly increases the protein expression of FLG and AQP3 in HaCaT cells [64]. In our study, schaftoside, an apigenin-di-C-glycoside, upregulated the expression of AQP3 and FLG. FLG and involucrin are two epidermal proteins that are downregulated in response to damage to the skin barrier. Additional insights from the literature reveal that apigenin and vicenin-1 (6,8-diglycosylflavone) increased the expression of FLG and claudin-1 upon exposure to UV radiation in a 3D skin model study [53]. Authors offer an explanation that FLG expression in normal human epidermal keratinocytes (NHEKs) was upregulated by activation of the aryl hydrocarbon receptor (AhR) [65]. In addition, apigenin exhibited aryl hydrocarbon receptor (AhR) antagonist activity in Caco2 colon cancer cells. Therefore, one explanation that can be inferred is that schaftoside may possibly upregulate the FLG expression in keratinocytes by modulating AhR. However, this hypothesis must be validated by additional studies in the future. Aesculin and schaftoside upregulated the epidermal differentiation marker, keratin-10 (K-10). K-1 and K-10 are structural keratin proteins responsible for hyperdifferentiation and differentiation of the epidermis, thereby regulating skin barrier function. A downregulation of these proteins normally occurs in conditions such as atopic dermatitis and epidermolytic hyperkeratosis [66,67].

There are many biophysical methods available for measuring skin hydration. The measurement of trans-epidermal water loss (TEWL) rates through human skin is the most used method in the cosmetics and skincare industry because TEWL directly correlates with skin barrier dysfunction. Ideally, TEWL should be as low as possible in healthy skin. Lower numbers correlate with less water loss, and high numbers correlate with increased water loss and poor barrier function [68]. *V. yedoensis* extract showed a hydration effect comparable to 5% glycerin in water. Further, *V. yedoensis* extract decreases the trans-epidermal water loss. In normal conditions, with an intact skin barrier, TEWL on human skin is on the order of 5–10 g/m^2^. Trans-epidermal water loss gives us the measure of the barrier function of the skin. The flux of water vapor that crosses the skin surface gives a measure of trans-epidermal water loss. As the water loss across the SC increases, the humidity next to the skin surface rises above ambient humidity [69]. This creates a humidity gradient above the skin surface that is proportional to the SC water loss. It characterizes the barrier function of the stratum corneum and thus allows the studying of the influence of physical or chemical factors on the integrity of the skin barrier. The more the skin barrier is perturbed, the higher the TEWL [70].

Primary intercellular barrier lipids, including cholesterol, essential fatty acids (EFAs), and ceramides, work together to form a barrier to prevent TEWL. Barrier lipids help maintain the skin’s natural collagen, elastin, and proteins by minimizing TEWL, increasing hydration and reinforcing the protective barrier [71]. In the DSC experiment, it can be observed that the intracellular lamellar lipids in the SC provide a tight and effective barrier to the passage of water through the tissue. Lipid disordering on the molecular level is translated into a deterioration of the barrier properties of the SC at the tissue level. A decrease of ≥2 °C is considered significant. *V. yedoensis* extract showed an equivalent effect on water vapor flux in lipid perturbation compared to 5% glycerin in water. Thus, this extract maintained the intact skin barrier properties.

To summarize, the phytochemical analysis and structural characterization of secondary metabolites within the *Viola yedoensis* extract led to the discovery of 14 new bioactive components, notably belonging to megastigmane glycosides and triterpene saponins. This marks the first report of their occurrence and distribution within *Viola yedoensis*. A comprehensive understanding of their presence necessitates an examination of their biosynthetic pathways within the plant. Therefore, to establish a biosynthetic relationship between the newly identified megastigmane glycosides, roseoside and eriojaposide B, and the corresponding calenduloside, we decided to elaborate on the related biosynthetic pathways envisioned to generate the apocarotenoids. Roseoside and eriojaposide B are megastigmane glycosides with a 3-oxo-α-ionol skeleton, while platanionoside B has a 3-hydroxy-β-ionol skeleton. Megastigmanes belong to the family of apocarotenoids. The C13 norisoprenoids possess a structure representative of carotenoid cleavage products. Both sesquiterpenes and carotenoids share a common biosynthetic pathway originating from β-Ionone (Figure 9) [72]. Calendulosides are glycosylated triterpene saponins reported mainly from the genus Calendula. The biosynthetic precursor for these oleanane-type pentacyclic triterpenoids is β-Amyrin (Figure 10). The first step in the biosynthesis of triterpenoid saponins involves the cyclization of 2,3-oxidosqualene to cycloartenol. This undergoes further cyclization to dammarenyl cation, which undergoes rearrangement to oleanane-β-Amyrin and finally to oleanolic acid [73].

## 4. Materials and Methods

### 4.1. Plant Material, Solvents, Chemicals and Standards

*Viola yedoensis* Makino was collected from Bozhou, Anhui province, China, in December 2019. The plant was identified by Dr. P. Santhan, and a voucher specimen of the entire plant has also been deposited at the Herbarium facility at L’Oreal (Advanced Research, Bangalore), India, with the voucher specimen number Loreal R & I/17-06-2020. They were stored under frozen conditions at −20 °C and protected from light. To determine the moisture content and proportions of organic and inorganic fractions, a thermogravimetric investigation was carried out using a muffle furnace (Nabertherm GmbH, Lilienthal, Germany). The experimental protocol involved dehydration at 100 °C for 12 h, followed by calcination at 650 °C for 4 h. Aesculin (CAS No: 531-75-9) (Sigma-Aldrich ≥ 98%) and schaftoside (CAS No: 51938-32-0) (Shangai Tauto Biotech ≥ 98%, Shangai, China) of pharmaceutical primary standards were procured from Merck and Sigma-Aldrich, respectively. All solvents and reagents utilized in this study were purchased from Sigma-Aldrich (St. Louis, MO, USA). HPLC-MS-grade acetonitrile and methanol were purchased from Merck (Lowe, NJ, USA). Laboratory-grade ethanol was obtained from Merck Millipore (Darmstadt, Germany). Milli-Q Integral 15 system (Merck Millipore, Burlington, MA, USA) was used to prepare the HPLC-grade water. Standard solutions were prepared at a concentration of 2.5 mg/mL in DMSO.

### 4.2. Flash Chromatography

Flash chromatography of the extract was performed using the Sepacore Flash system X50, Buchi with a C18 column (250 mm × 2.5 mm). A total of 100 mg of the sample was solubilized in MeOH:Water (50:50, 5 mL) and filtered by 0.45 µm (Millipore Millex-HN). A gradient elution with ethanol:water mobile phase at the flow rate of 15 mL/min led to the preparation of a specific *Viola yedoensis* extract composition (Appendix A).

### 4.3. LC/HRMS/MS Instrumentation and Conditions

Acquisitions were realized on an Orbitrap Fusion high-resolution mass spectrometer (HRMS) equipped with an electrospray source (ESI), an Ultimate 3000 ultra-high-performance liquid chromatography (UPLC) system, and a photodiode-array detector (PDA) from Thermo Fisher Scientific. Separation was achieved on an ACQUITY BEH Shield RP18 (Waters, 100 × 2.1 mm, 1.8 μm) at 35 °C. Mobile phases used were water and acetonitrile (ACN), both acidified with 0.1% of formic acid. The gradient used was as follows: 0–1 min: 1% ACN, 1–15 min: 1 to 100% ACN, 15–20 min: 100% ACN, 20.1–25 min: 1% ACN, at a flow rate of 0.6 mL/min (direct hyphenation with HRMS). The autosampler was set at 10 °C. The injection volume was 2 μL. The wavelength range of the PDA detector was 210–700 nm. Acquisitions were performed on scan range *m/z* 100–1100 operated separately in positive (+3.8 kV) and negative (−3.4 kV) ion modes. ESI source parameters were set as follows: vaporizer temperature: 400 °C, capillary transfer tube temperature: 325 °C, sheath gas pressure: 30 arbitrary units, auxiliary gas: 10 arbitrary units, and sweep gas: 2 arbitrary units. The RF Lens was set at 60%, and AGC Target was in standard mode, with an injection time of 100 ms. Analyses were performed using full scan mode, and full scan MS—data-dependent MS2 mode (ddMS2). Collision experiments based on non-resonant excitation mode higher-energy collisional dissociation (HCD at NCE = 20%) and sequential MSn experiments based on resonant excitation mode (collision-induced dissociation, CID, at NCE = 25%) experiments were acquired. The mass spectrometer was calibrated prior to analysis with Pierce^TM^ FlexMix^TM^ calibration solution (Thermo Fisher Scientific, reference A39239). The mass resolution power of the analyzer was 120,000 m/Δm, full width at half maximum (FWHM) at *m/z* 200, for singly charged ions. All the data were manually inspected using Thermo Fisher Scientific software: Freestyle module of Xcalibur (version 4.4), Compound Discoverer (version 3.3) in combination with product ion spectral database mzCloud (2024 update) and an in-house mzVault database.

To estimate the difference in gas-phase acidity of OH groups between a dehydrated hexose and a phenolic aglycone, model molecules’ values were used. The ΔH°_(acid)(M)_ of model molecules M are considered as (i) tetrahydro-2-pyranol, 1540 kJ/mol or cyclohexanone, 1533 kJ/mol, used for a dehydrated hexose and (ii) phenol, 1462 kJ/mol or meta diphenol as resorcinol, 1450 kJ/mol used for an aglycone. Values obtained from the NIST WebBook.

### 4.4. Enzyme-Linked Immunosorbent (ELISA) Assays

The supernatant of extracted solution was injected into LC/HRMS system. Standard solutions were prepared at a concentration of 2.5 mg/mL in DMSO. The enzyme-linked immunosorbent (ELISA) assay Dulbecco’s modified Eagle’s medium (DMEM, optimized 1× DMEM with 4.5 g/L glucose, L-glutamine, sodium pyruvate, and HaCaT cell line) was purchased from AddexBio Technologies Inc., Krishgen. Human CD 44 ELISA kit was purchased from Abcam (Cambridge, UK), aquaporin-3 from Novus Biologicals, and keratin-10 and filaggrin from Abbexa (Cambridge, UK) and Cussabio (Houston, TX, USA). The cell lysis buffer (10×) was from Cell Signal Technology (Boston, MA, USA) and 6 well plates from Corning (New York, NY, USA). Envision 2105 Multimode Plate Reader from Perkin Elmer (Waltham, MA, USA) was used. TEWL and DSC analysis: PIE System, Falcom toolbox, 16 and 14 mm Die set, Punching Press, TEWL Sample holders, Rotary Shaker, ESPEC Temperature and Humidity Chamber, Mettler Toledo Analytical Balance, Circular Adhesive Tape. DSC: Mettler Toledo DSC 1 STAR e System (IN) (Mettler-Toledo India Private Limited, Mumbai, India) or DSC TA instrument Q200 (FR), Standard 40 µL Aluminum Crucible with Pin (IN) or Tzero Aluminium Hermetic Pans (FR).

### 4.5. Extraction and Fractionation of the Extract

The whole plant of *V. yedoensis* Makino (100 g) was dried, powdered, and subjected to hydroalcoholic extraction (50:50, EtOH:water, 65 °C, 3 cycles) to yield 30 g of the extract. The extract (30 g) was subjected to HP-20 resin fractionation [20:80 (EtOH:water), 60:40 (EtOH:water) and 100% EtOH] to yield three fractions VY-1, VY-2, and VY-3 (Figure 10). The fraction VY-1 and VY-2 comprised of the coumarins, aesculin, and esculetin, respectively. The fractions VY-3 rich in flavone glycosides was further fractionated by flash chromatography.

The fraction VY-3 was further fractionated by Flash chromatograph system. A total of 100 mg of the sample was solubilized in MeOH:Water (50:50, 5 mL) and filtered by 0.45 µm (Millipore Millex-HN). A gradient elution (Appendix A) with methanol:water mobile phase and C18 column at the flow rate of 15 mL/min led to the isolation of fractions *Viola yedoensis* extract (*V. yedoensis* extract) enriched with flavonoids. A fraction of *V. yedoensis* extract was subjected to HRMS analysis for the identification of the compounds. Compounds **1**–**15** were previously reported from the same plant. Compounds **16**–**29** are reported for the first time from the genus *Viola*.

### 4.6. Cell Viability Assay

The effect of the compounds on cell viability after 24 h treatment was tested. Cells (10,000 per well) were seeded in a 96-well opaque plate and incubated overnight at 5% CO_2_, 37 °C. The cells were treated with *V. yedoensis* extract 0.03%, aesculin 1%, schaftoside 0.1%, and sodium hyaluronate 5 kDa (positive control) 0.1% and incubated for 24 h at 5% CO_2_, 37 °C. Cell Titer Glo Reagent (Promega-G7571) was added to the plate, and incubated, and luminescence was measured in Envision plate reader.

### 4.7. ELISA Assay

HaCaT cell line was trypsinized and seeded at the density of 1 × 10^6^ cells/well in a 6-well plate and kept for overnight incubation at 37 °C, 5% CO_2_ conditions. The following day, cells were treated with the following test compounds: *V. yedoensis* extract at 2%, aesculin at 1%, and hyaluronic acid at 0.1%, which were weighed and solubilized in culture media. The plant extract (0.03%) was further diluted at 1:1 ratio in DMEM to achieve test concentration. The test compounds were sterile-filtered with 0.2 µm syringe filter. The media were aspirated, and 1 mL/well of the prepared filtered dilutions was added to the 6-well plate as per the plate map and kept for overnight incubation at 37 °C, 5% CO_2_ conditions. Following 24 h treatment with the test compounds, cells were harvested. The supernatant (spent media) was collected and centrifuged at 10,000 rpm for 10 min. The pellet containing the debris was discarded, and the supernatant was stored at −80 °C until further use. For the lysate, the adherent cells were trypsinized, the pellet was washed twice with PBS, and 50 µL of 1× cell lysis buffer was added to the cell pellet. The cells were incubated on ice for 30 min with intermittent mixing and stored at −80 °C for complete lysis. On the day of the experiment (ELISA), the frozen lysates were thawed on ice, centrifuged at 13,000 rpm at 4 °C for 15 min. The supernatant was collected and used for the ELISA.

The ELISA kits use the sandwich-ELISA principle. The micro-ELISA plates provided in the kits have been pre-coated with an antibody specific to target protein. Standards or samples are added to the micro-ELISA plate wells and combined with the specific antibody. Then, a biotinylated detection antibody specific for target protein and avidin-horseradish peroxidase (HRP) conjugate are added successively to each micro-plate well and incubated. Free components are washed away. The substrate solution is added to each well. Only those wells that contain target protein, biotinylated detection antibody, and avidin-HRP conjugate will appear blue in color. The enzyme–substrate reaction is terminated by the addition of stop solution, and the color turns yellow. The optical density (OD) is measured spectrophotometrically at a wavelength of 450 nm ± 2 nm. The OD value is proportional to the concentration of the target protein. The concentration of the target protein in the samples is calculated by comparing the OD of the samples to the standard curve. Data analysis: The average of duplicate readings for each standard and samples was taken. The average values were subtracted with the average zero standard optical density.

### 4.8. TEWL Measurements

The TEWL measurements were recorded in vitro on isolated stratum corneum using a motorized assembly. The stratum corneum samples were placed in the temperature and humidity control chamber for equilibrium at 30 °C and 30% RH. After overnight equilibrium, TEWL measurements “T1” were recorded. Measurement of flow of water crossing a sample of SC situated above a water tank was made in a glove box kept at 30 °C and 30% RH by means of a hygrometric probe coupled to a robotic sample holder and a data acquisition system Derma lab. The hygrometric probe (Appendix A) contains two humidity sensors located at different heights in the cylindrical probe chamber so that the humidity gradient can be measured and used to calculate the flux of water vapor. The measurements were performed before and after treatment for each sample across two different batches of stratum corneum with 4 replicates on each batch. Assessment of intercellular stratum corneum lipid perturbation by DSC: After TEWL measurements, the same samples were used for the DSC analysis after equilibrating them for an additional 4 h at 75% RH, 25 °C. The measurements were performed using aluminum hermetic-sealed pans and heating from 10 °C to 110 °C at a rate of 5 °C/min.

## 5. Conclusions

Our investigation aimed to screen a new composition of *Viola yedoensis* extract for its ability to restore epidermal barrier function through the expression of key epidermal protein markers, alongside enhanced skin hydration properties. In a primary screen for hydration benefits, the *V. yedoensis* extract was found to increase the protein expression levels of CD44 and aquaporin-3 (AQP3) in HaCaT cells by sandwich-ELISA assay. Owing to this new biological efficacy, we applied an integrated LC-HRMS/MS-based metabolomics approach under resonant (CID 25%) and non-resonant collisional (HCD 20%) excitation conditions, together with state-of-the-art computational tools, to provide acute insights regarding the characterization of bioactive components from the aerial parts of *V. yedoensis*. This methodology involved data analysis pipelines of untargeted metabolomics analysis that enabled the characterization and identification of 29 secondary metabolites, 14 of which are reported here for the first time in *V. yedoensis*, namely, esculetin, aesculin, apigenin, kaempferol C-glycosides, megastigmane glycosides, roseoside, platanionoside B, and eriojaposide B isomer, together with the rare calenduloside F and esculetin diglucoside, which are reported for the first time from the genus *Viola*.

Additionally, the study demonstrated for the first time that *V. yedoensis* extract, enriched with flavone glycosides, can be cosmetically employed for restoring the epidermal barrier function through the expression of key epidermal protein markers, CD44 and AQP3, with enhanced skin hydration properties. Furthermore, two major secondary metabolites, aesculin and schaftoside, were discovered to be the main components of *V. yedoensis* to upregulate the expression of the epidermal differentiation marker, K-10, which is chiefly responsible for maintaining the structural and mechanical integrity of the stratum corneum. The apigenin-C-glycoside, schaftoside, significantly increased the protein expression level of CD44, while the coumarin glycoside, aesculin, increased the AQP3 levels and upregulated the expression of keratin-10 protein, comparable to that of the positive reference, sodium hyaluronate. It is interesting to note that schaftoside significantly upregulated the expression of filaggrin, thereby suggesting that schaftoside could be a useful natural bioactive component in the future treatment of some of the most common dermatological diseases, such as atopic dermatitis and psoriasis. A comprehensive structure–activity study of the flavone C-glycosides and coumarin glycosides has yet to be reported, and it would be relevant in the future to examine more analogues from this class in this context. Most importantly, our results provide encouragement that the study of known glycosides from the *Viola* genus could lead to the discovery of additional novel bioactive components with enhanced skin barrier function and hydration benefits.

## Figures and Tables

**Figure 1 ijms-25-12770-f001:**
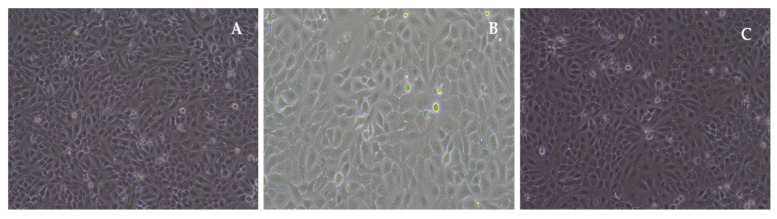
Representative images (10× magnification) of HaCaT cell line. (**A**) Untreated, (**B**) treated with *V. yedoensis* extract (0.03%), and (**C**) positive control (sodium hyaluronate 0.1%).

**Figure 2 ijms-25-12770-f002:**
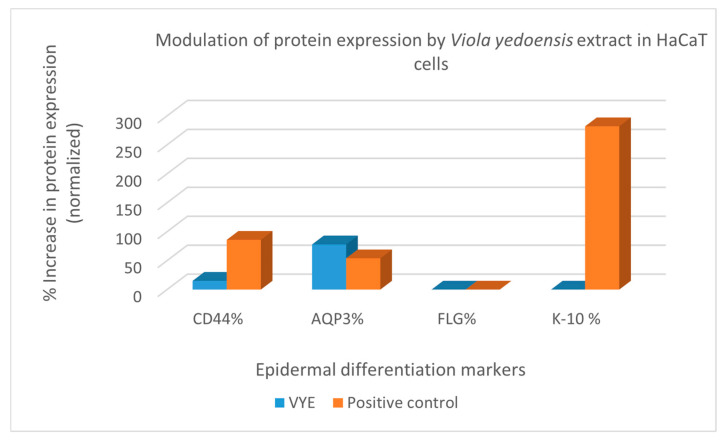
Regulation of epidermal differentiation markers—CD44, AQP3, and K-10—in HaCaT cells treated by *V. yedoensis* extract and positive control (sodium hyaluronate, 0.1%). Values are average of experiments in duplicate.

**Figure 3 ijms-25-12770-f003:**
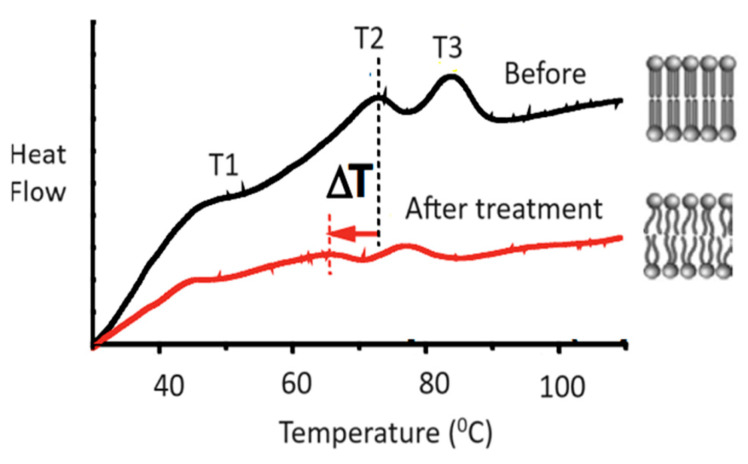
Endothermic peaks in DSC with *V. yedoensis* extract.

**Figure 4 ijms-25-12770-f004:**
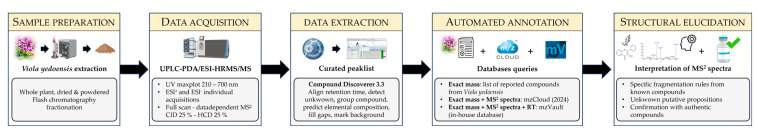
Methodology used to annotate secondary metabolites of *V. yedoensis* extract, according to recommendations from the Chemical Analysis Working Group (CAWG) [25].

**Figure 5 ijms-25-12770-f005:**
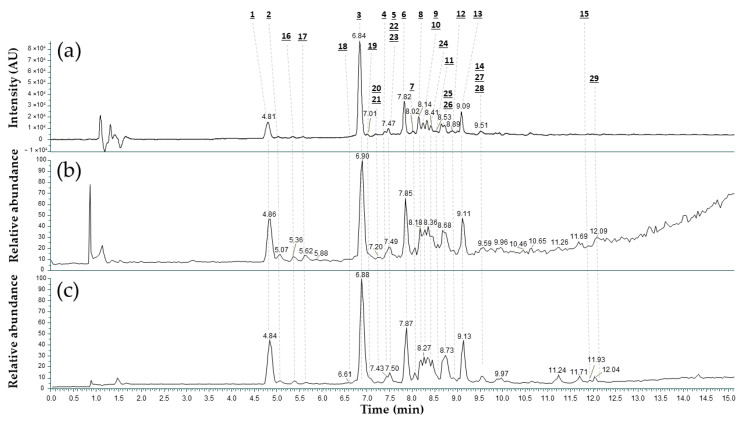
LC-UV/HRMS chromatograms of *V. yedoensis* extract. (**a**) UV maxplot 210–700 nm, TIC with (**b**) ESI positive and (**c**) ESI negative, *m/z* 100–1100.

**Figure 6 ijms-25-12770-f006:**
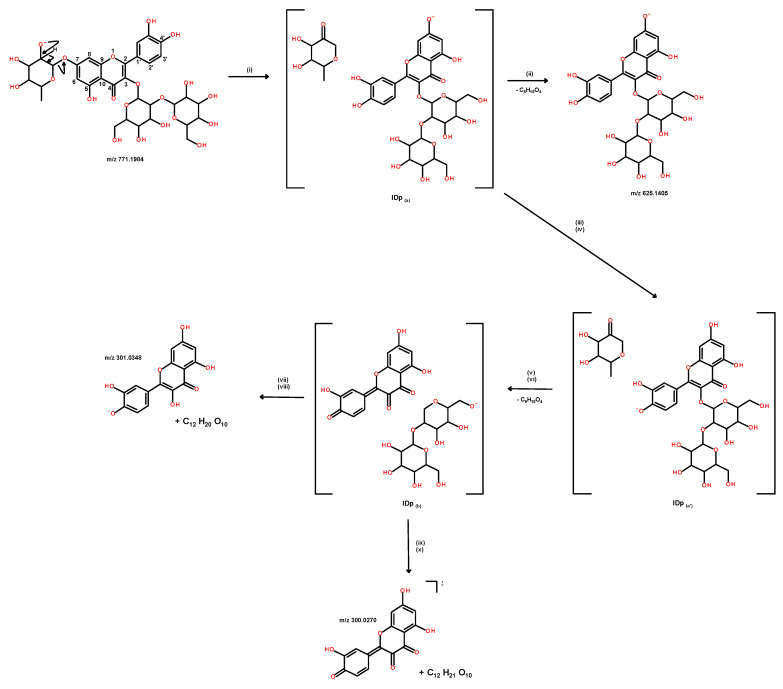
Proposed mechanisms of stepwise gas-phase fragmentations of collisionally activated deprotonated glycoconjugates, desorbed as a various deprotomers provided in LC-ESI/HRMS/MS from *V. yedoensis* extract. Main product ions displayed in collision spectrum of quercetin-3-sophoroside-7-O-rhamnoside deprotomer mixture are used as model for description of possible interpretation of various product ion: (i) stepwise mechanisms via molecular anion isomerization to ion/dipole (IDp_(a)_) based on 1,2-hydride transfer from a secondary hydroxyl (or, eventually, the primary hydroxyl) at the rhamnose moiety promoting the C-O linkage cleavage at C(7), (ii) direct splitting of IDp(a) yielding *m/z* 625 by the dehydrated rhamnose neutral loss (C_6_H_10_O_4_, 146 Da), or (iii) internally to IDp_(a)_, deprotonation of dehydrated rhamnose by proton transfer to phenoxy at C(7), (iv) intermediate used for long-distance exothermic negative charge transfer from C_6_H_9_O_4_^−^ to phenol at C(4′) (intermediate IDp_(a’)_) as well as at other phenolic C(5) and C(3′) positions (but ineffective for the further reactions), (v) followed by C_6_H_10_O_4_ loss, and (vi) resulting intermediate isomerization into IDp_(b)_ induced by the charge migration from phenoxy at C(4′) to dehydrated sophoroside (C_12_H_21_O_10_) by cleavage of C-O linkage at C(3), and (vii) internal IDp_(b)_ hydride transfer from C_12_H_21_O_10_^−^ to aglycon followed by (viii) release of dehydrated C_12_H_20_O_10_ sophoroside (324 Da) to yield deprotonated aglycon m/z 301; alternatively from IDp_(b)_ (ix) internal electron transfer from C_12_H_21_O_10_^−^ to aglycon and (x) formation of oxidized aglycon radical anion *m/z* 300 by the C_12_H_21_O_10_^•^ radical release. Note the sequences from (v) to (viii) and (v) to (x) can occur initially from *m/z* 771 to give rise to formation of *m/z* 447 and *m/z* 446 product ions, respectively, possibly yielding formation of *m/z* 301 and *m/z* 300 by release of dehydrated rhamnose and its radical form.

**Figure 7 ijms-25-12770-f007:**
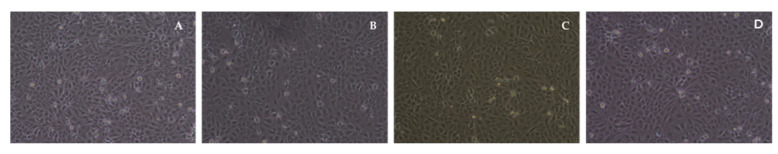
Representative images (10× magnification) of HaCaT cell line. (**A**) Untreated, (**B**) aesculin (1%), (**C**) schaftoside (0.1%), and (**D**) positive control (sod. hyaluronate 0.1%).

**Figure 8 ijms-25-12770-f008:**
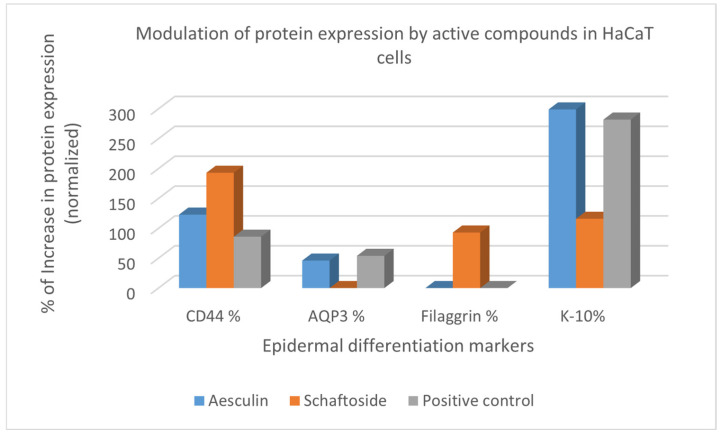
Regulation of epidermal differentiation markers CD44, AQP3, FLG, and K-10 in HaCaT cells treated by aesculin (1%), schaftoside (0.1%), and positive control (sod. hyaluronate 0.1%). Values are average of experiments in duplicates.

**Figure 9 ijms-25-12770-f009:**
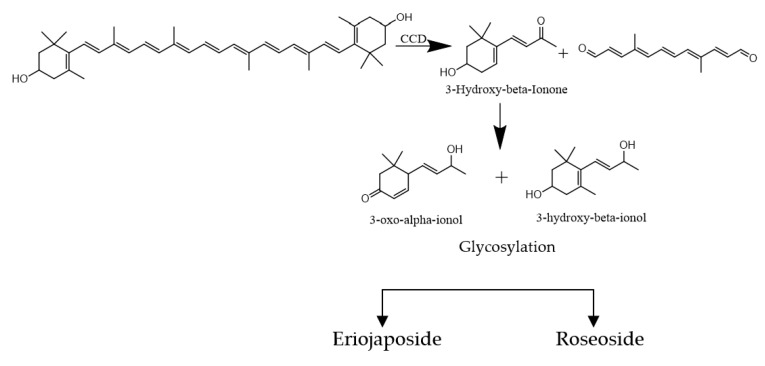
Biosynthetic pathway of roseoside and eriojaposide B.

**Figure 10 ijms-25-12770-f010:**
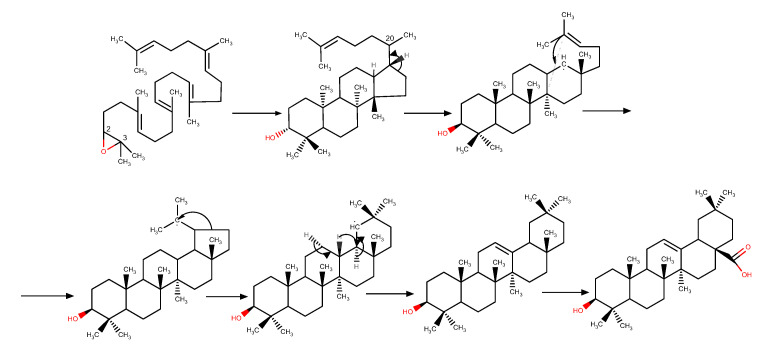
Biosynthetic pathway of calenduloside.

**Table 1 ijms-25-12770-t001:** TEWL and DSC analysis of *V. yedoensis* extract.

Analysis Parameters	*V. yedoensis* Extract	Water	5% Glycerine in Water
TEWL (water flux ΔTEWL, %)	±5.2%	±5%	±5.5%
DSC (lipid perturbation ΔTm °C)	±0.30 °C	±2 °C	±0.11 °C

**Table 2 ijms-25-12770-t002:** Reported compounds in *V. yedoensis* identified in extract with *m/z* of respective precursor [M − H]^─^ ions and their product ions using LC/ESI-HRMS/MS under non-resonant collisional excitation conditions (HCD 20%).

Index	Rt (min)	Name/CAS Number (Identified with ^a^ MS² Spectra, ^b^ RT, ^c^ De Novo)	Glycosyl Linkage to Aglycon Backbone	Precursor Ion *m/z*	δ ppm	Elemental Composition	Product Ion *m/z* (‡ Oxidation)	References
1	4.75	Cichoriin ^c^/531-58-8	O-coumarin	339.0726	3.1	C_15_H_16_O_9_	177.0188, 133.0294, 105.0350	[28]
2	5.17	Aesculin (Esculin) ^a,b^/531-75-9	O-coumarin	339.0725	2.9	C_15_H_16_O_9_	177.0189, 133.0294, 105.0347	[29]
3	6.85	Esculetin ^a,b^/305-01-1	Coumarin	177.0188	0.1	C_9_H_6_O_4_	133.0295, 105.0348	[29]
4	7.40	Vicenin 2 ^a,b^/23666-13-9	Di-C-flavone	593.1519	2.1	C_27_H_30_O_15_	503.1203, 473.1092, 383.0778, 353.0672	[30]
5	7.61	Isocarlinoside ^c^/83151-90-0	Di-C-flavone	579.1361	1.9	C_26_H_28_O_15_	489.1044, 459.0936, 429.0830, 399.0725, 369.0620	[31]
6	7.84	Schaftoside ^a,b^/51938-32-0	Di-C-flavone	563.1412	2.0	C_26_H_28_O_14_	473.1094, 443.0990, 383.0778, 353.0672	[31]
7	8.02	Prionanthoside ^c^/161842-81-5	O-coumarin	381.0831	2.4	C_17_H_18_O_10_	177.0189	[23]
8	8.13	Isoorientin ^a,b^/4261-42-1	C-flavone	447.0937	2.1	C_21_H_20_O_11_	4290829, 357.0621, 327.0515	[31]
9	8.21	Apigenin 6-C-α-l-arabinopyranosyl-8-C-β-d-xylopyranoside ^c^/677021-30-6	Di-C-flavone	533.1307	2.2	C_25_H_26_O_13_	515.1203, 473.1094, 443.0987, 383.0782, 353.0673	[31]
10	8.24	Euphorbetin ^c^/35897-99-5	Coumarin dimer	353.0309	3.3	C_18_H_10_O_8_	335.0204, 309.0409, 177.0188	[32]
11	8.49	Isovitexin ^a,b^/38953-85-4	C-flavone	431.0988	2.3	C_21_H_20_O_10_	3410669, 311.0566, 283.0617	[33]
12	8.85	Cynaroside (luteolin 7-O-glucoside) ^a,b^/5373-11-5	O-flavone	447.0938	2.4	C_21_H_20_O_11_	327.0525, 285.0407 ^‡^	[32]
13	9.14	Quercetin 3-glucoside ^a,b^/482-35-9	O-flavonol	463.0887	2.3	C_21_H_20_O_12_	300.0278 ^‡^, 271.0254, 255.0299, 151.0033	[29]
14	9.51	Astragalin (kaempferol 3-O glucoside) ^a,b^/480-10-4	O-flavonol	447.0938	2.4	C_21_H_20_O_11_	357.0630, 327.0515, 284.0330 ^‡^, 255.0300, 227.0354, 151.0037	[30]
15	11.86	Luteolin ^a,b^/491-70-3	Flavone	285.0408	3.1	C_15_H_10_O_6_	175.0397, 151.0034	[34]

Identified with authentic compound: ^a^ MS^2^ spectrum, ^b^ RT, and ^c^ de novo structural elucidation. ^‡^ Oxidation of aglycone to quinone-like structure characterized by high electron affinity yielding preferentially radical anion by electron transfer from deprotonated carbohydrate in ion/dipole complex (vide infra).

## Data Availability

Data is contained within the article and Appendix A.

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
