# Peer review of "Improved Skin Barrier Function Along with Hydration Benefits of Viola yedoensis Extract, Aesculin, and Schaftoside and LC-HRMS/MS Dereplication of Its Bio-Active Components"

_ijms, 2024, doi:10.3390/ijms252312770_

Round 1
Reviewer 1 Report
Comments and Suggestions for Authors
The article “Restoration of Epidermal Barrier with Improved Skin Hydration benefits of Viola yedoensis extract, Aesculin, Schaftoside and LC-HRMS/MS dereplication of its Bio-active components” is devoted to the study of the composition of the alcohol extract as well as its effect on skin hydration. In general, the article is well structured, with interesting results on the composition of the extract and possible ways of biosynthesis of its individual bioactive molecules.
However, some comments have been made to the article:
The article's title does not quite reflect the essence of the work. The authors did not investigate the restoration of the skin's epidermal barrier, so I recommend correcting the title.
The article makes it unclear how the authors obtained aesculin and shaftoside for their research. The “Materials and Methods” section should describe this in detail.
It should specify how the amount of expressed proteins was converted to the total amount of cell protein since the results are presented in % (Figs. 2, 8). These figures should be improved by specifying the labels. The error bars should be indicated when presenting the ELISA results in Figures 2 and 8.
The authors should indicate which statistical methods of analysis were used and which criteria were applied since, in the article, the authors mention statistically significant differences between the studied parameters.
In paragraph 4.4, the authors mention that they used an involucrin assay kit, but such data are not available in the text of the article.
There are inaccuracies in the text of the article; for example, the authors write that “These keratins are integral components composed of proline-rich proteins such as SPRR, Involucrin, loricrin and filaggrin [5]”, but involucrin, loricrin, and filaggrin are not keratins.
The text of the article contains unexplained abbreviations and inaccuracies in their spelling (e.g., PI3K or P13K, line 182). The plant's species name should be written in italics (lines 530, 537, 570, 571, 588, 777).
The authors should pay attention to inconsistencies in the conclusion. The authors write that V. yedoensis extract increased the expression of CD44, AQP3, Keratin-10 (K-10), and Filaggrin (FLG) proteins in HaCaT cells, which contradicts the obtained results.
In general, the references should be brought in line with the requirements, and the spelling of the authors' names should be checked (for example, lines 910 and 954). It is also desirable to include more recent literature.
Author Response
Please find attached file for our responses to reviewers comments.

Reviewer 2 Report
Comments and Suggestions for Authors
The research article discussed the "Restoration of Epidermal Barrier with Improved Skin Hydra- 2 tion benefits of Viola yedoensis extract, Aesculin, Schaftoside 3 and LC-HRMS/MS dereplication of its Bio-active components".
This is well organized and presented. However, I have several considerations:
(1) What does the "V. yedoensis" species belong to? Is this a medicinal plant? Will there exist toxicity?
(2) Could you explain the mechanisms of V. yedoensis on the skin in detail?
(3) What is the purpose of using TEWL analysis in the Skin Hydration evaluation?
Author Response
V. yedoensis belongs to the Violaceae family. In traditional oriental medicine used for the treatment of inflammation-related diseases including swelling, sores, boils, furuncles, carbuncles, snakebites, and acute and chronic hepatitis.
Using an ELISA assay, we evaluated the protein expression levels of CD44, Aquaporin-3 (AQP3), Filaggrin, and Keratin-10. V. yedoensis extract upregulated the levels of CD44 and AQP3 by 15% and 78%, respectively. Additionally, V. yedoensis extract demonstrated a comparative effect on water vapor flux in TEWL and lipid perturbation in DSC versus the reference, glycerin.
Transepidermal water loss (TEWL) is a measurement that assesses the skin's barrier function and hydration.